# Diagnostic Tool for the Identification of *Bactrocera dorsalis* (Hendel) (Diptera: Tephritidae) Using Real-Time PCR

**DOI:** 10.3390/insects15010044

**Published:** 2024-01-08

**Authors:** Domenico Rizzo, Claudia Gabriela Zubieta, Patrizia Sacchetti, Andrea Marrucci, Fortuna Miele, Roberta Ascolese, Francesco Nugnes, Umberto Bernardo

**Affiliations:** 1Laboratory of Phytopathological Diagnostics and Molecular Biology, Plant Protection Service of Tuscany, Via Ciliegiole 99, 51100 Pistoia, Italy; domenico.rizzo@regione.toscana.it (D.R.); claudiagabriela.zubieta@regione.toscana.it (C.G.Z.); andrea.marrucci93@gmail.com (A.M.); 2Department of Agriculture, Food, Environment and Forestry (DAGRI), University of Florence, Piazzale delle Cascine, 18, 50144 Florence, Italy; patrizia.sacchetti@unifi.it; 3Department of Agricultural Food and Agro-Environmental Sciences, University of Pisa, Via del Borghetto 80, 56124 Pisa, Italy; 4Institute for Sustainable Plant Protection (IPSP), National Research Council (CNR), P.le Enrico Fermi 1, 80055 Portici, Italy; fortuna.miele@ipsp.cnr.it (F.M.); roberta.ascolese@ipsp.cnr.it (R.A.); francesco.nugnes@ipsp.cnr.it (F.N.); 5Department of Biology, University of Naples Federico II, Via Vicinale Cupa Cintia 21, 80126 Napoli, Italy

**Keywords:** oriental fruit fly, phytosanitary survey, priority pest, prompt diagnosis, quarantine insect pest, real-time PCR

## Abstract

**Simple Summary:**

The accurate identification of the Oriental fruit fly, *Bactrocera dorsalis*, is complicated by its similarities to other species and taxonomic uncertainties. This represents a significant threat to fruit crops as it is already present in Europe, and this is a cause for great concern. To expedite identification, a reliable method using a unique technical approach was developed. The initial phase involved collecting specimens from the population present in Italy to create a large and representative sample, enabling us to optimize the method. This method has demonstrated high sensitivity and accuracy in detecting small amounts of *B. dorsalis* DNA. It now serves as a valuable tool for routine diagnostics, facilitating efficient pest management and detection. Given the recent infestations in Italy, this diagnostic protocol is crucial for monitoring and preventing the passive spread of *B. dorsalis* in Europe.

**Abstract:**

Accurate identification of *Bactrocera dorsalis* (Hendel) (Diptera: Tephritidae), commonly known as the Oriental fruit fly, is a significant challenge due to the morphological convergence and taxonomic uncertainties of species belonging to the same genus. This highly polyphagous species poses a significant threat to fruit crops. With its potential establishment in Europe becoming a growing concern, there is an urgent need for rapid and efficient diagnostic methods. The study presented here introduces a diagnostic protocol based on real-time PCR using a TaqMan probe for the early and reproducible identification of *B. dorsalis*. Specimens representing the genetic diversity of the Italian population were collected and analyzed. Specific primers and probe were designed based on the conserved regions and an in silico analysis confirmed their specificity. The assay conditions were optimized, and analytical sensitivity, specificity, repeatability, and reproducibility were evaluated. The protocol showed high sensitivity and specificity, accurately detecting low DNA concentrations of *B. dorsalis*. This standardized method provides a reliable tool for routine diagnostics, enhancing the accuracy and efficiency of identifying the Oriental fruit fly at all stages of its development, thereby facilitating effective pest management measures. The development of this diagnostic protocol is crucial for monitoring and supporting efforts to prevent the passive spread of *B. dorsalis* in Europe, particularly in light of the recent active infestations detected in Italy.

## 1. Introduction

*Bactrocera dorsalis* (Hendel) (Diptera: Tephritidae), commonly known as the Oriental fruit fly, belongs to a monophyletic group consisting of twelve closely related species known as the *B. dorsalis* complex [1,2,3]. The identification of specimens within the species complex is very challenging due to morphological convergence and overlapping characteristics. Additionally, taxonomic uncertainties and recent changes in synonymies further complicate accurate identification [4,5,6,7]. As a matter of fact, the taxonomic classification of the *B. dorsalis* complex has been subjected to and still undergoes continuous revisions [2,3,8,9,10], resulting in provisional identifications and indeterminations regarding certain species and taxa [4,5,11]. Furthermore, the presence of shared COI haplotypes among species of the genus *Bactrocera* adds complexity to ultimate identification [3,9,12].

Moreover, the diagnostic instability of *B. dorsalis*, combined with its morphological similarity to other species, presents challenges in accurately determining its true host range. Identification difficulties impede the confident association of scientific papers with a specific species in nearly all cases. Despite previous investigations by [1,9,13], there remains a gap in our understanding on this matter. Precise identification is crucial for comprehending its biological traits and evaluating its effects [5].

The Oriental fruit fly is a highly polyphagous insect pest that can produce damage to over 400 fruit crop species [2,14,15]. A recent analysis of the potential distribution of *B. dorsalis* has highlighted its status as a significant threat and assumes that it has the necessary potential to establish populations in extensive regions of southern Europe [4]. Previous studies have acknowledged the potential for *B. dorsalis* to spread across the continents, although they presented less alarming scenarios for Europe [16]. 

Nonetheless, *B. dorsalis* has demonstrated the ability to overwinter in areas with climatic conditions like those of temperate regions in North America and Europe [17,18]. Consequently, the possibility of its establishment in territories with a mild climate is plausible, reinforced by the remarkable adaptability displayed by the species [19]. From the native Indo-Asian region, the Oriental fruit fly hit Africa and quickly invaded almost the whole continent [20]. Likewise, it could be further conveyed and spread to new areas through the fruit trade. Due to the huge economic danger caused by the possible accidental introduction of this fruit fly, the species is categorized as a quarantine pest in all the countries at risk of entry [15].

In Europe, *B. dorsalis*, along with other non-European Tephritidae species, is classified as a union quarantine pest according to Annex II, Part A of Commission Implementing Regulation [21]. Additionally, it is designated as a priority pest according to [22]. Consequently, the monitoring of *B. dorsalis* is an essential part of the mandatory phytosanitary activities conducted by European countries. This monitoring activity involves visual checks of fruits, analysis of infestation, and the placement of hundreds of traps, each baited with at least one of two attractants (methyl eugenol or torula).

The monitoring efforts conducted in several countries have revealed frequent detections of *B. dorsalis* in Europe, specifically in Italy, Austria, and France [23]. However, until mid-2022, all recorded occurrences of the species were classified as “simple incursions”, indicating the absence of established populations in these areas [12,23,24,25,26].

A significant shift in the situation occurred in June 2022 with the detection of the first active infestations of *B. dorsalis* in Italy [26,27]. These findings prompted the Campania Region to develop an action plan aimed at eradicating or at least mitigating the settlement of *B. dorsalis*. One of the measures required by the plan was the restriction of the movement of fruit from the demarcated area to prevent the pest spreading [28].

As previously mentioned, the reliable identification of *B. dorsalis* presents difficulties due to diagnostic limitations.

This is further supported by the lack of officially recognized diagnostic tools shared at the international level, apart from those primarily reliant on morphological identification being conducted by specialized taxonomists [5]. Therefore, there is an urgent need to develop a rapid and effective diagnostic method for the ultimate identification of *B. dorsalis*, particularly in relation to the population or species present in Italy. This need is particularly critical given the potential spread of the species in Europe. 

The morphological identification of the preimaginal stages of *B. dorsalis* poses even greater challenges than adult identification. Although a taxonomic key is available to assist in identifying the mature larvae of frequently intercepted tephritids at entry points in Europe, this key is characterized by several constraints [29,30]. These shortcomings encompass the overlapping ranges of variation among different characters and the omission of accounting for the variability between the right and left mandibular sclerites [29]. Furthermore, recent reports have highlighted the extensive morphological variability in the mandibular sclerites in *Ceratitis capitata* (Wiedemann) (Diptera: Tephritidae), which can make the morphological identification of tephritid larvae based on this characteristic impossible [31]. 

Given the complexities and limitations of the current protocols for identifying the various life stages of *B. dorsalis*, it is imperative to develop a method that can be effectively applied to fruit inspection and survey activities at points of entry. Such a method should enhance the accuracy and efficiency of identifying the Oriental fruit fly and facilitate effective pest management measures.

The aim of this study was to develop a diagnostic protocol for the diagnosis of preimaginal stages and adults of *B. dorsalis*. This protocol was based on real-time PCR using TaqMan probe technology and was validated with additional non-target specimens.

## 2. Materials and Methods

### 2.1. Insect Collection 

In 2022, as part of a field survey conducted in the Campania Region, adults of *B. dorsalis* were collected using traps that were baited with methyl-eugenol (ME). The specimens collected from the Italian population were specifically selected to represent the genetic variability observed in Campania. To ensure a comprehensive representation, individuals were chosen from various locations, at different dates, and showing different mitochondrial haplotypes. 

Furthermore, adults of *Bactrocera latifrons* (Hendel), the solanum fruit fly, collected in 2019 were used for the purpose of comparison [25]. Finally, larvae were collected in the field from the areas where active infestations were discovered, specifically in Campania. The additional non-target specimens used in the validation tests (as shown in Table 1) were collected during territorial surveys or obtained from scientific institutions according to collaborative agreements. The identification of these specimens was performed using both morphological analysis with the appropriate identification keys and barcoding analysis targeting the COI gene [32]. These specimens were preserved in 70% ethanol at −20 °C for storage. 

The fruit fly specimens collected from infested fruits or from traps were identified utilizing the available taxonomic keys [8,14,33,34,35]. 

Table 1 provides a comprehensive list of both the target and non-target species used in this research. The selection of non-target species was based on two criteria: their taxonomic relatedness to the target species and unrelated species that share the same host plants and exhibit frugivorous behavior.

### 2.2. DNA Extraction

Duplicate DNA extractions were performed on both the adult and larval specimens using 4 mL of 2% CTAB buffer, following the procedure outlined in [36]. The subsequent steps, including DNA purification, quantification, and assessment of the contamination levels, were conducted according to the methodology described in [37]. The elution of DNA was carried out in 100 µL of nuclease-free distilled water and utilized immediately for the qPCR reactions or stored at −20 °C until further use.

The DNA extraction from the *B. dorsalis* and *B. latifrons* samples, provided by the IPSP laboratory, was conducted using a Chelex–proteinase K protocol as described by [12,25]. For low-quality samples, the protocol was modified by extracting the DNA from whole insect bodies in a solution containing 200 µL of Chelex and 10 µL of proteinase K, incubated at 55 °C for 24 h. For all the samples, the DNA dilutions were normalized to a concentration of 5 ng/µL and subsequently subjected to real-time PCR analysis using a dual-labeled probe targeting a highly conserved region of the 18S rDNA, as described by [38]. DNA amplifiability tests were conducted to assess the quality of the extractions and detect the presence of any potential inhibitors.

### 2.3. Design of the Primers and Probe for the qPCR Assays 

The primers and the qPCR dual-labeled probe used (Table 2) were designed following the methodology detailed in [37]. The design process was based on the conserved Cytochrome Oxidase I, yet variable regions of *B. dorsalis* were obtained from GenBank.

The in silico specificity analysis for both the probe and primers was conducted following the methodology described in [37]. The results of the in silico analyses indicated that the assays under analysis showed no significant matches or extended sequences with the non-target organisms in the database (Appendix A). The optimal primer annealing temperatures for qPCR amplification were determined based on the evaluation conducted in [37]. A temperature gradient ranging from 55 °C to 62 °C was applied, and the primer and probe concentrations varied from 0.2 to 0.5 µM. To assess the specificity of the amplification, two tubes containing 2 μL of nuclease-free distilled water were included as No-Template Controls (NTCs) in each run for the probe protocols. All the real-time gene amplification reactions were conducted using a CFX96 thermal cycler (Bio-Rad, Hercules, CA, USA). Measurement of the fluorescence, along with the automatic definition of the fluorescence threshold to identify the inflection points (indicative of increasing kinetics) and, consequently, any positivity, was performed using the CFX Maestro ver. 2.3 software (Bio-Rad). In selecting the reference sequences for assay construction, various factors were considered, including the potential presence of diagnostic polymorphisms, the utilization of specific mitochondrial genomic regions, and highly conserved genes that offer interspecific variability concurrently [39]. To assess the in silico specificity of the chosen sequences, the BLAST^®^ software (Basic Local Alignment Search Tool; 2.14 version, http://www.ncbi.nlm.nih.gov/blast (accessed on 1 September 2023)) was employed. The most closely related nucleotide sequences were identified using the expected amplicon of the real-time PCR protocol with a probe as the query and were aligned using MAFFT [40], implemented in the Geneious^®^ 10.2.6 software (Biomatters, http://www.geneious.com (accessed on 1 September 2023)). Alignments related to the assays under consideration were performed by distinguishing inclusivity and exclusivity [41]. Inclusivity entailed comparing the sequences related to *B. dorsalis* from various geographical origins at a global scale with the reference sequence (Appendix A), while exclusivity implied testing the highest number of non-target *Bactrocera* spp. to highlight the specificity of the assay (Appendix A). In addition, positive and negative amplification controls were included for the target samples. The primers and probe were synthesized by Eurofins Genomics (Eurofins Genomics, Ebersberg, Germany).

### 2.4. Validation Method for the qPCR Probe 

To ensure the test’s eligibility as a standardized method for routine diagnostics, all the relevant characteristics, including analytical sensitivity, analytical specificity, repeatability, and reproducibility, were determined according to the criteria established in [41,42]. Accomplishing these recognized standards guarantees the reliability, comparability, and acceptance of the test as a standardized method in routine diagnostics within the EPPO region.

The analytical sensitivity of the qPCR probe test, aimed at determining the limit of detection (LoD), was evaluated using DNA from a single sample at a concentration of 10 ng/µL (starting concentration). Serial dilutions of the DNA were prepared in triplicate at a ratio of 1:5. The repeatability and reproducibility were assessed by testing 8 samples in triplicate, conducted in two separate series following the methodology described in [43]. The evaluation range in the analytical sensitivity assessment included dilutions between 2 ng/µL and 25.6 fg/µL (based on 2 µL DNA per sample). This allowed for the effective evaluation of the technique’s performance in detecting low concentrations of the target DNA.

## 3. Results

### 3.1. DNA Extraction 

In Table 3, the average DNA concentrations (ng/µL) are provided for the extracted DNA from the *B. dorsalis* adults and larvae, along with the corresponding standard deviations (SD). The absorbance ratios (A260/280) indicate the purity of the extracted DNA. Additionally, the Cq values obtained in the qPCR amplification targeting the 18S ribosomal gene represent the amplifiability of the DNA samples.

### 3.2. Assay Conditions of the TaqMan Probe Protocol 

For the TaqMan probe qPCR, the optimal reaction mix consisted of 10 µL of 2× QuantiNova PCR Master Mix probe (QIAGEN, Hilden, Germany). The concentrations of the primers and probe were set at 0.4 µM and 0.2 µM, respectively. The optimal thermal conditions were as follows: an initial denaturation step at 95 °C for 2 min, followed by 40 cycles of denaturation at 95 °C for 10 s, and annealing/extension at 55 °C for 40 s. The average Cq values obtained using this assay starting from a diluted concentration up to 10 ng/µL were equal to 21.5 ± 0.36 for the DNA extracts of *B. dorsalis* adults, while for the larvae, there were values equal to 17.5 ± 0.26 (Appendix A). Samples were considered positive upon the identification of a distinct inflection point in the real-time PCR curves, accompanied by a discernible increase in kinetics. Despite the absence of non-specificity or kinetic anomalies during the initial 40 cycles of the analytical specificity tests, a prudent recommendation is made to adopt Cq values below 35 as the designated threshold (Figure 1).

### 3.3. Validation of the Proposed Methods 

The test developed in this study demonstrated inclusivity for *B. dorsalis*, meaning it successfully detected and identified this target species. Additionally, the test exhibited exclusivity, accurately discriminating against the non-target organisms that were tested. The analytical sensitivity (LoD) was determined to be 0.128 pg/µL (Table 4).

The linearity value (R^2^) obtained for the qPCR assay was 0.98792, indicating a robust correlation between the DNA concentration and the corresponding Cq values (Figure 2). 

In the qPCR probe protocol, the values were just over 24 both for repeatability and reproducibility (Table 5). Most of the values obtained for the repeatability and reproducibility fall below the suitable parameters of variability (<0.5 SD) [44], indicating that the assays can be considered repeatable and reproducible.

## 4. Discussion

The current *B. dorsalis* infestation in Italy, with the ongoing detection of new adults in Italy and Europe, emphasizes the urgency of rapid and reliable identification methods. In particular, distinguishing *B. dorsalis* from *C. capitata* is crucial to preventing the spread of infested fruits. The movement of fruits infested by *B. dorsalis* poses significant risks, while *C. capitata*, already widespread in Europe, carries minor consequences, even if caution is crucial because it is essential to also limit the spread of highly virulent insect haplotypes due to human activities [45,46].

In this study, a qPCR assay based on a TaqMan probe has been developed for screening and diagnostic confirmation. This technique has the capability to overcome the considerable genetic variability in *B. dorsalis* [19]. In fact, it has been demonstrated, both in silico and in vivo, to accurately identify all the tested *B. dorsalis* specimens, showing the different haplotypes present in the Campania Region in 2022.

Moreover, obtaining reliable results from insect matrices is often difficult, making the choice of DNA extraction method crucial for achieving accurate diagnostic outcomes. In this study, two extraction protocols were utilized, both of which have previously demonstrated their efficacy in research involving various insect pests and a diverse array of environmental samples, such as frass, exuviae, and wood chips [36,47]. It is worth noting that these protocols employ entirely different approaches. Despite their methodological differences, both protocols produced results that aligned with the study’s expectations.

Thus, analyzing the parameters used to define qualitative and quantitative states (concentration and A260/280), the results were in line with what was expected for this type of test.

The proposed method incorporates the TaqMan probe technology, which exhibits both advantages and disadvantages when compared to alternative techniques like qPCR with SybrGreen. The probe-based test, in particular, is more expensive and intricate in terms of its design and planning compared to the SYBR Green qPCR method. The latter, however, offers the benefit of a simpler design and execution, along with cost-effectiveness. It should be noted that these advantages sometimes result in reduced specificity within qPCR assays employing SYBR Green, as documented in prior studies [48,49]. Test reliability is right now of the utmost importance because, despite the implementation of various measures and eradication attempts in the infested areas (such as the Campania Region in Italy), the population currently present in Italy certainly represents the greatest potential threat to the entire European territory [26,28].

Therefore, rapid and accurate identification is important when handling fruit consignments coming from infested areas, but it becomes crucial when it is presumed that the fruits originate from free orchards or locations near the infested areas.

The proposed qPCR probe technique, both in screening mode and in the case of confirmation of positive suspects, allows samples to be identified with certainty and faster than with sequencing. This ensures the prompt implementation of effective risk mitigation measures. In particular, it will allow phytosanitary inspectors to prevent the export of infested fruits from affected areas and swiftly detect the presence of *B. dorsalis* in tephritid-infested fruits during import into pest-free regions, such as European countries.

Several interception methods have been used in the past. However, for *B. dorsalis*, only four methods are available, including a costly LAMP assay [50] and a PCR with specific primers capable of distinguishing only 10 species within the genus *Bactrocera* [51]. It is essential to consider that real-time LAMP assays frequently incur high reagent costs. Additionally, the experimental designs of LAMP assays may, at times, lead to assays with lower sensitivity or specificity compared to a qPCR assay. Moreover, a multiplex PCR assay [52] and a single-gene TaqMan real-time PCR [53] were developed and assessed for specific species or, in the case of [54], exclusively for *B. zonata*.

However, unlike previous works, the proposed qPCR probe method outlined in this study adopts a much more comprehensive approach, encompassing a wider range of tephritid species, including those belonging to the genera *Anastrepha* and *B. zonata*. The results, both in silico (for over 200 *Bactrocera* spp.) and in vivo, displayed favorable results when compared to similar frugivorous pests and demonstrated an analytical sensitivity (LoD) of 0.128 pg/µL, denoting a limit of detection in line with similar work [49].

The qPCR probe assay successfully detected all previously identified haplotypes of *B. dorsalis* using sequence analysis, demonstrating its versatility and diagnostic accuracy. The in silico tests, including the inclusivity and exclusivity libraries, confirmed that the test can cover potential haplotypes and genetic variability, even in the face of the genetic diversity within the *B. dorsalis* complex. The high repeatability and reproducibility (100%) with variability generally below the 0.5% threshold confirm the test’s reliability and suitability for routine diagnostics, especially in areas where the presence of *B. dorsalis* needs verification. Additionally, the test aligns with [41], further endorsing its potential as a standardized diagnostic method.

## 5. Conclusions

The development of a diagnostic protocol using real-time PCR and a TaqMan probe provides a promising approach to the early and reproducible identification of *B. dorsalis*. This protocol addresses the challenges associated with accurate identification and can enhance the accuracy and efficiency of identifying the Oriental fruit fly, facilitating effective pest management measures. The validation of the protocol demonstrates its eligibility as a standardized method for routine diagnostics, ensuring reliability and comparability.

## Figures and Tables

**Figure 1 insects-15-00044-f001:**
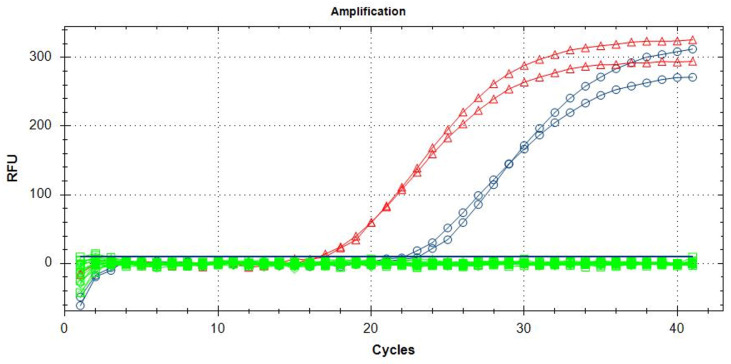
Amplification curves relating to larvae (red triangles) and adults (blue circles) of *B. dorsalis* in addition to the non-targets (green squares) listed in Table 1.

**Figure 2 insects-15-00044-f002:**
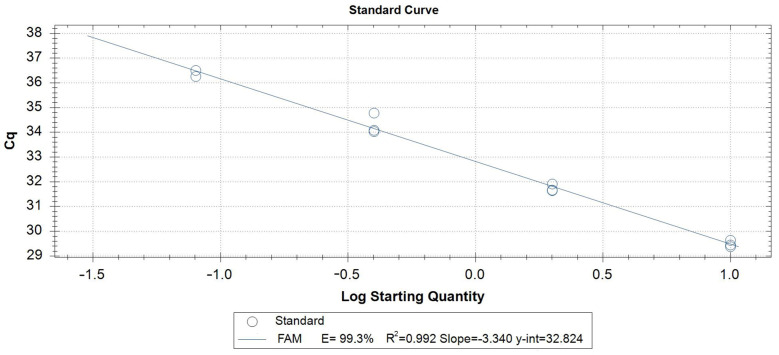
Standard curves relating to the qPCR probe assay using serial dilutions (1:5) of *B. dorsalis* DNA.

**Table 1 insects-15-00044-t001:** Insect samples (including both target and non-target specimens) used in this study to assess the analytical specificity (inclusivity and exclusivity) of the test. Abbreviations: UoF = University of Florence, UoP = University of Pisa, IPSP = Institute for Sustainable Plant Protection, CNR, PPS-T = Plant Protection Service, Tuscany. XSIT = XSIT Ltd. mass-rearing facility (Citrusdal, Western Cape, South Africa).

Species	Sample Code	Life Stage	Geographical Origin of Samples	Supplier
*Anastrepha fraterculus* (Wiedemann, 1830)	MR 000752	Adult	Ecuador	UoF
	MR 000795	Adult	Ecuador	UoF
*Anastrepha leptozona* (Hendel, 1914)	MR 000817	Adult	Ecuador	UoF
	MR 000823	Adult	Ecuador	UoF
	MR 001658	Adult	Ecuador	UoF
*Anastrepha ludens* (Loew, 1873)	MR 000753	Adult	USA	UoF
	MR 000820	Adult	USA	UoF
	MR 001659	Adult	USA	UoF
	MR 001710	Adult	USA	UoF
	MR 001721	Adult	USA	UoF
*Anastrepha obliqua* (Macquart, 1835)	MR 000821	Adult	Ecuador	UoF
	MR 001715	Adult	Ecuador	UoF
*Anastrepha serpentina* (Wiedemann, 1830)	MR 000283	Adult	USA	UoF
	MR 000822	Adult	USA	UoF
	MR 001701	Adult	USA	UoF
	MR 001689	Adult	USA	UoF
	MR 001722	Adult	USA	UoF
*Bactrocera dorsalis* (Hendel, 1912)	MR 001712	Adult	Italy	UoF
	MR 001720	Adult	Italy	IPSP
	MR 001709	Adult	Italy	IPSP
	MR 001718	Adult	Italy	IPSP
	MR 001716	Adult	Italy	IPSP
	MR 000814	Adult	Italy	IPSP
	MR 001563	Adult	Italy	IPSP
	MR 001713	Adult	Italy	IPSP
	MR 001638	Adult	Italy	IPSP
	MR 001714	Adult	Italy	IPSP
	MR 001717	Adult	Italy	IPSP
	MR 001711	Larva	Italy	IPSP
	MR 001719	Larva	Italy	IPSP
	MR 000801	Larva	Italy	IPSP
	MR 001683	Larva	Italy	IPSP
	MR 001684	Larva	Italy	IPSP
	MR 000239	Larva	Italy	IPSP
	MR 001697	Larva	Italy	IPSP
	MR 001747	Larva	Italy	IPSP
	MR 000764	Larva	Italy	IPSP
*Bactrocera latifrons* (Hendel, 1915)	MR 000752	Adult	Italy	IPSP
	MR 000823	Adult	Italy	IPSP
*Bactrocera oleae* (Rossi, 1790)	MR 001658	Adult	Italy	UoF
*Bactrocera zonata* (Saunders, 1842)	MR 000753	Adult	Afghanistan	UoF
*Ceratitis capitata* (Wiedemann, 1824)	MR 000820	Adult	Italy	UoF
	MR 000821	Larva	Italy	UoF
	MR 000283	Adult	Italy	IPSP
	MR 000822	Larva	Italy	PPS-T
*Cydia pomonella* (Linnaeus, 1758)	MR 000259	Adult	Italy	UoF
	MR 000790	Adult	Italy	UoF
*Grapholita* (*Aspila*) *molesta* (Busck, 1916)	MR 001617	Adult	Italy	UoF
*Rhagoletis cerasi* (Linnaeus, 1758)	MR 001618	Adult	Italy	UoF
*Rhagoletis completa* (Cresson, 1929)	MR 001648	Larva	Italy	UoP
	MR 001619	Adult	Italy	UoF
*Thaumatotibia leucotreta* (Meyrick, 1913)	MR 001678	Adult	South Africa	XSIT
	MR 001620	Adult	South Africa	XSIT
	MR 001621	Larvae	Interception at the port of Leghorn (Italy)	PPS-T

**Table 2 insects-15-00044-t002:** List of the primers and probe designed for *B. dorsalis*.

Name	Sequence	Size (bp)	Reference Sequence
Bdors_235F	CACCAGTCATATTGTGAG	105	OP056621.1
Bdors_340R	GTGTCATGAAGAATAATATCTAC		
Bdors_308P	FAM-AGCAAGAACTACTCCTGTTAATCCTCC-BHQ1		

**Table 3 insects-15-00044-t003:** Average concentrations of the extracted DNA (± SD), absorbance ratio (A260/280), and Cq values of 18S [38] for the assayed samples.

Sample	DNA Conc (ng/µL) ± SD	A260/280 Ratio	Cq (18S)
Adult	25.0 ± 2.68	1.9 ± 0.12	18.4 ± 0.81
Larva	203.2 ± 90.50	1.8 ± 0.08	16.2 ± 0.14

**Table 4 insects-15-00044-t004:** Analytical sensitivity (LoD) of tests using serial dilutions (Cq means ± SD). Cq values above 35 were considered negative results.

Dilutions 1:5	qPCR Probe
10.0 ng/µL	17.5 ± 0.26
2.0 ng/µL	20.4 ± 0.19
0.4 ng/µL	22.1 ± 0.12
0.08 ng/µL	23.9 ± 0.20
0.016 ng/µL	27.4 ± 0.13
3.2 pg/µL	28.3 ± 0.51
0.64 pg/µL	30.2 ± 0.13
0.128 pg/µL	33.0 ± 0.09
25.6 fg/µL	-

**Table 5 insects-15-00044-t005:** Repeatability and reproducibility values (mean ± SD) obtained using TaqMan probe protocol at an initial concentration of 0.04 ng/µL on extracted DNA.

Sample	qPCR Probe Test	
	Repeatability	Reproducibility
1	24.5 ± 0.66	24.9 ± 0.40
2	24.5 ± 0.62	24.5 ± 0.20
3	24.1 ± 0.25	24.4 ± 0.15
4	24.2 ± 0.31	24.3 ± 0.22
5	24.3 ± 0.04	24.6 ± 0.35
6	24.4 ± 0.40	24.3 ± 0.17
7	24.4 ± 0.27	24.2 ± 0.21
8	24.5 ± 0.41	24.5 ± 0.19

## Data Availability

Almost all the data will be available upon the publication of the manuscript as Appendix A. Furthermore, all other data are deposited into the Institute for Sustainable Plant Protection—National Research Council (IPSP-CNR), P.le E. Fermi, 1-80055 Portici (NA), Italy, and are available on request to Umberto Bernardo, umberto.bernardo@ipsp.cnr.it.

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
