# Peer review of "Diagnostic Tool for the Identification of Bactrocera dorsalis (Hendel) (Diptera: Tephritidae) Using Real-Time PCR"

_insects, 2024, doi:10.3390/insects15010044_

Round 1

Reviewer 1 Report

Comments and Suggestions for Authors

The research article entitled “Development of a diagnostic tool for the early and reproducible identification of Bactrocera dorsalis (Hendel) (Diptera Tephritidae) using Real-Time PCR with TaqMan probe technology” describes the development and validation of a novel TaqMan assay for the detection of an EU invasive alien species (oriental fruit fly). The methodological approach is scientifically sound and includes all necessary steps for the validation of the assay from design to analytical and diagnostic: specificity/sensitivity etc and its application can help the quick identification of this agriculturally important species. I have the following comments to make:

(1)  Introduction, page 2, lines 59-60: Please mention here which are the challenges specifically and elaborate more on the diagnostic limitations.

(2)  Introduction: It would be helpful to briefly present the current distribution of this species, and which are the means of invasion apart from fruit trade (if any).

(3)  Introduction: How is monitoring for the invasion by this species currently performed and which are the points of entry?

(4)  Methods / Results: It is not entirely clear how the SYBR Green approach was used, and which are the results- how do these compare to the TaqMan assay- these should be presented and the comparison of the two methods to be discussed.

(5)  Results: A Table with the Cq values for the target and non-target species is needed; for the latter if not detected at all, also do mention.

(6)  Discussion, lines 252-254: Not clear, please rephrase.

(7)  Discussion, line 292: to be fair cost is not the issue with LAMP, maybe a criticism on specificity/sensitivity?

Author Response

Dear referee,

please find my point-by-point response attached.

All the best

Umberto

Reviewer 2 Report

Comments and Suggestions for Authors

The manuscript describes a new assay to distinguish B. dorsalis from other closely related species with the B. dorsalis complex that can be used at points of entry. The authors claim they have developed a tool that is reproducible, species-specific, and has high sensitivity.

While some data is shown, there is no data or figure to support the claim that this assay works. The authors need to provide information on the assay, how it works, what program was used, what thresh hold of fluorescence was used to determine a positive sample, what measurement of fluorescence was used, etc. It may be useful to include a spreadsheet in the supplemental documents that shows each sample, the "value" used to determine if it was B. dorsalis, this same value at different DNA quantities, etc. 

I'm also concerned with the limited sampling. All the B. dorsalis used were collected in Italy - the same region? different populations? More information is needed. It seems that samples outside of Italy should also be included. Where are the flies most likely coming from that enter Italy? These populations should be included. 

Other minor issues:

Line 138 define ME at first use (I think line 120).

Line 136 - close the parenthesis

Line 303 - QPCR should be qPCR

Author Response

Dear referee,

please find point-by-point response attached.

All the best

Umberto

Round 2

Reviewer 2 Report

Comments and Suggestions for Authors

The authors have made changes that address my concerns. I feel it is ready for publication.

Author Response

Many thanks